# The Evolution of Glycoside Hydrolase Family 1 in Insects Related to Their Adaptation to Plant Utilization

**DOI:** 10.3390/insects13090786

**Published:** 2022-08-30

**Authors:** Shulin He, Bin Jiang, Amrita Chakraborty, Guozhi Yu

**Affiliations:** 1College of Life Science, Chongqing Normal University, Chongqing 401331, China; 2College of Life Science, Anhui Normal University, Beijing Rd. 1, Wuhu 241000, China; 3Faculty of Forestry and Wood Sciences, Czech University of Life Sciences Prague, Kamýcká 129, 16500 Prague, Czech Republic; 4College of Life Science, Sichuan Agricultural University, Xinkang Rd. 46, Ya’an 625014, China

**Keywords:** β-glucosidase, duplication and loss, tandem duplication, neofunctionalization, plant cell wall, ecology

## Abstract

**Simple Summary:**

β-glucosidase is a crucial enzyme in the adaptation of insects to plant cell wall digestion and plant metabolite detoxication. However, the evolution of this enzyme remains unclear. Here, we fill this gap by investigating the evolution of insect glycoside hydrolase family GH1, accounting for the majority of members of β-glucosidase. We found the GH1 gene family to be present in all insect species with large gene numbers in insects directly feeding on plant cell wall components. Furthermore, the large gene numbers were associated with a complex evolutionary history, including tandem duplication and neofunctionalization. These results highlight the evolutionary traits of an important insect enzyme and provide further insights into our understanding of the evolution of insect–plant interactions.

**Abstract:**

Insects closely interact with plants with multiple genes involved in their interactions. β-glucosidase, constituted mainly by glycoside hydrolase family 1 (GH1), is a crucial enzyme in insects to digest plant cell walls and defend against natural enemies with sequestered plant metabolites. To gain more insights into the role of this enzyme in plant–insect interactions, we analyzed the evolutionary history of the GH1 gene family with publicly available insect genomes. We found that GH1 is widely present in insects, while the gene numbers are significantly higher in insect herbivores directly feeding on plant cell walls than in other insects. After reconciling the insect GH1 gene tree with a species tree, we found that the patterns of duplication and loss of GH1 genes differ among insect orders, which may be associated with the evolution of their ecology. Furthermore, the majority of insects’ GH1 genes were tandem-duplicated and subsequently went through neofunctionalization. This study shows the evolutionary history of an important gene family GH1 in insects and facilitates our understanding of the evolution of insect–plant interactions.

## 1. Introduction

Insects, one of the most speciose animal groups on earth, adapted to various ecosystems and closely interact with many other groups, especially plants [1]. Most insects are herbivorous and utilize plants as food resources by digesting plant cell walls with a series of digestive enzymes, such as mannanases, endo-β-1,4-glucanases, and β-glucosidases [2]. They also adapt to various plant defense mechanisms, such as the production of enzymes for detoxification of plant toxic metabolites [3] and the sequestration of metabolites to form a similar dual-component defense system to that in plants [4,5], where toxic secondary metabolites are activated by plant enzymes upon tissue damage [6]. The molecular mechanisms of plant cell wall digestion and adaptations to plant defense are revealing [2,3,7]; however, the evolution of genes related to these mechanisms remains unclear.

β-glucosidases, enzymes that hydrolyze glycosidic bonds to release non-reducing terminal glucosyl residues from glycosides and oligosaccharides, play significant roles in the utilization of plants by insects [2,5,6,8]. The characterized β-glucosidases are mainly involved in cellobiose digestion [9,10,11,12], the breakdown of glucosinolates sequestered from host plants to form a dual-component defense system [5,6,13,14,15,16,17,18,19,20,21], and communication and recognition among sexual or social interactions [22,23,24,25]. Moreover, these functions have been found in different groups of insects [9,10,11,12,22,26,27,28,29,30,31].

Glycoside hydrolase 1 family (GH1) presents a substantial part of the characterized β-glucosidases and is widely present in bacteria, insects, vertebrates, and plants [8,32]. In prokaryotes, GH1s are divided into eight subfamilies, including six in bacteria and two in archaea [32]. In eukaryotes, GH1s are categorized into six subfamilies, including one in plants, one in insects, three in mammals, and one in fungus.

GH1s are encoded by a moderate-size gene family in insects. It has been suggested that the insect GH1s diverged from the same animal GH1 gene ancestor and further evolved with the functions in plant nutrient utilization [8], which may be related to the evolution of plants as a derived food source in insects [33]. In addition, the myrosinase activity of GH1 has been indicated to be a newly gained function; an aphid myrosinase shows a different evolutionary origin in comparison with plant myrosinases [27] but a similar global structure to other β-glucosidases in the same species [34]. Moreover, previous studies indicated that myrosinase has evolved independently in aphids and beetles [18]. However, a systematic evolutionary analysis of GH1 genes spanning various insect orders is lacking, hindering our understanding of the role of GH1 genes in insect adaptations to plant utilization.

Here, we used publicly available genome resources to investigate the evolution of the insect GH1 gene family. We identified GH1 genes from selected insect species and a few other arthropod species and compared the gene number in different orders and feeding groups. Subsequently, we built a gene phylogeny of GH1 genes based on protein sequences and inferred the duplication and loss of this gene family by reconciling the gene tree with a species tree. Finally, we analyzed the duplication modes of GH1 genes and syntenies of GH1 in a set of selected species. Our analysis revealed the evolutionary history of insect GH1 genes, including tandem duplication and loss as well as neofunctionalization. Along with the diverse enzyme activities of insect GH1s, the evolution of the GH1 gene family provides further insights into our understanding of insect–plant interactions.

## 2. Materials and Methods

### 2.1. Data Collection

In this study, we took advantage of available genomes, mainly RefSeq genomes from the NCBI genome database [35]. The public genome assemblies with annotations of 54 arthropod species, spreading over the main lineages of the insect group, were downloaded (accessed on 25 April 2021, Figure 1, Appendix A). The completeness of the genome assemblies was determined by using BUSCO with the insecta_odb10 dataset [36]. We extracted the longest isoform for each gene based on the annotation files and proteomes for subsequent analyses.

### 2.2. Glycoside Hydrolase Family 1 Identification

The corresponding proteomes were used for GH1 identification by following the annotation steps of a standalone tool run_dbcan2, which is based on the Carbohydrate-Active enZYmes Database and runs 3 different prediction tools: HMMER, Diamond, and Hotpep [37]. The prediction suite can provide a high annotation accuracy with the combination of outputs from two prediction tools; therefore, proteins predicted as GH1s from 2 tools were considered confident GH1s.

To identify the origin of predicted genes, the proteins were queried against the non-redundant (nr) protein database in NCBI, and the last common ancestor (LCA) of up to the top 10 best targets for each query was inferred using the ete3 toolkit [38]. The corresponding gene was considered as microbial origin if the LCA of the protein was lower than Metazoa. Subsequently, to separate contaminations from horizontal gene transfers, the gene structures and blast results of the microbial origin genes were manually checked. The genes from bacteria were removed for further analysis.

### 2.3. Gene Tree Inference

We followed the alignment steps of AQUA (Muller et al. 2010), with a few modifications. Briefly, the GH1 protein sequences were aligned with 3 different alignment tools, MAFFT [39], MUSCLE [40], and Clustal Omega [41], and the alignments were refined with RASCAL [42]. Assessed with NORMD [43], the alignment with the highest NORMD score selected from the pre-refined and post-refined alignments was used for phylogeny inference. The phylogeny was built with IQTREE [44] with model selection [45] and 1000 ultrafast bootstrap [46].

### 2.4. Gene Duplication and Loss Inference

As a dated phylogeny for all the species used in this study is not available, we used Notung to infer gene duplication and loss, which requires a non-dated species tree and a gene tree [47]. The non-dated species tree with the species in this study was compiled from a few studies [48,49,50,51,52,53] considering the relative positions of species. Gene duplication and loss were inferred with the duplication–loss event model; the rearranged model was implemented for reconciliation with a 90% bootstrap threshold to minimize the penalty of low-supported branches [54].

### 2.5. Duplicate Mode Inference and Collinearity Analysis

Both duplicate mode inference and collinearity analysis were performed in MCScanx [55] for the selected species of the following four insect orders separately: Hymenoptera, Coleoptera, Lepidoptera, and Diptera. The selected species were: *Agrilus planipennis*, *Anoplophora glabripennis*, *Onthophagus taurus,* and *Tribolium castaneum* from Coleoptera; *Aedes aegypti*, *Anopheles gambiae*, *Culex quinquefasciatus,* and *Drosophila melanogaster* from Diptera; *Apis mellifera*, *Nasonia vitripennis*, *Polistes dominula*, *Atta colombica*, and *Bombus terrestris* from Hymenoptera; *Danaus plexippus*, *Galleria mellonella*, *Manduca sexta*, *Spodoptera frugiperda*, and *Bombyx mori* from Lepidoptera. Most of the species have chromosomal-level genome assembly, except *A. planipennis*, *A. glabripennis*, and *O. taurus* in Coleoptera; *G. mellonella* in Lepidoptera; and *P. dominula* and *A. colombica* in Hymenoptera. The collinearity analysis was conducted with MCScanx, and the output was plotted with circos [56]. The duplication mode of genes in each species was inferred by using duplicate_gene_classifier.

### 2.6. Statistical Analysis

To determine the effect of feeding groups on insect GH1 gene evolution, we compared the gene numbers in different dietary groups by using phylANOVA from phytools. As phylANOVA requires a dated species tree, a phylogeny from a previous study [57] was used: Only the species or genus presented in the phylogeny were analyzed. Additionally, the herbivorous groups were further separated into plant-cell-wall feeding and non-plant-cell-wall feeding for comparison. The Holm–Bonferroni method was used to adjust the pairwise comparisons. In addition, the Pearson correlation test was performed between BUSCO scores of species at the tips and duplications/losses inferred at the corresponding terminal branches in the reconciliation output.

## 3. Results

### 3.1. GH1s in Insects and Other Arthropods

Most of the identified genes originate from metazoans except three genes: two in *Varroa jacobsoni* (LOC111273514 and LOC111273528) and one in *Acyrthosiphon pisum* (LOC100569078). After individually checking the genes in the NCBI database, we found that all three genes were located at unplaced scaffolds with one intron in LOC111273528 and no intron in LOC111273514 and LOC100569078; furthermore, the blast targets, except themselves, were from bacteria. Therefore, we considered these genes as contaminants from bacteria and excluded them from subsequent analyses.

Subsequently, we summed up the GH1 gene numbers in all the 54 arthropod species observed (Figure 1). We found that species from Chelicerata and *Penaeus vannamei* in crustaceans had no GH1 genes, whereas *Daphnia magna* had five GH1 genes. In the sister group of all insects, *Folsomia candida*, 17 copies of GH1 were determined. In insects, the gene numbers differed among orders, with large numbers in Blattodea, Lepidoptera, and Coleoptera, and small numbers in Diptera, Hymenoptera, and hemipteroid insects (Figure 2). In Blattodea, the GH1 gene number ranged from 7 in *Zootermopsis nevadensis* to 12 in both *Blattella germanica* and *Cryptotermes secundus* (Figure 1). In hemipteroid insects, only one copy of GH1 was found in bugs, *Diaphorina citri* and *Cimex lectularius*, and in lice, *Pediculus humanus*. By contrast, relatively large numbers of GH1 were found in plant-feeding species in this group, namely 14 copies in thrips, *Frankliniella occidentalis*, and 8–11 copies in aphids, *Aphis gossypii* and *A. pisum*. In Hymenoptera, the GH1 gene number varied from 5–8 in wasps to 1–4 in ants, bees, and sawflies. Noteworthily, a large number of GH1 genes were found in beetles as well as in butterflies and moths (Figure 1): In beetles, the number ranged from 6 in *Nicrophorus vespilloides* to 53 in *Anoplophora glabripennis*; in butterflies, the GH1 gene number varied from 18 in *Papilio machaon* to 32 in *Bicyclus anynana*, which was similar to that in moths ranging from 12 in *Amyelois transitella* to 35 in *Manduca sexta*. In Diptera, we found 6–8 GH1 gene copies in mosquitoes and 1–4 GH1 gene copies in flies.

### 3.2. GH1 Gene Numbers Related to Their Feeding Behaviors

As the functions of GH1 are related to plant cell wall digestion, we compared the GH1 gene numbers among different feeding groups (Figure 3). The gene numbers in herbivores showed no significant difference (F = 3.59, *p* = 0.117), compared with the gene numbers in non-herbivores. However, when separating herbivores into two groups depending on whether they feed on plant cell walls, we found the GH1 gene numbers were significantly related to their feeding behaviors (F = 16.71, *p* = 0.002). Herbivores directly feeding on plant cell walls, including wood- and leaf-feeding species, had significantly larger GH1 gene numbers than herbivores not directly feeding on plant cell walls (t = 5.15, *p* = 0.006), such as phloem sap or cell content, and non-herbivores (t = 4.82, *p* = 0.006). In comparison, the gene numbers in herbivores not feeding on plant cell walls were not significantly different from those found in non-herbivores (t = 0.87, *p* = 0.516).

### 3.3. Phylogenetic Tree of GH1 Genes in Insects

The phylogeny of GH1 was built with protein sequences of identified genes from selected species. According to the phylogeny, the insect GH1 family could be separated into six groups (Group I–VI) (Figure 4). Group I contained the most diverse GH1s from most insect orders except Hymenoptera (Figure 4a). Group II consisted of genes from Lepidoptera and Diptera; both Group III and Group IV were composed of genes only from Lepidoptera (Figure 4b). Group V comprised the second most diverse GH1s from Hymenoptera, Hemipteroid, Lepidoptera, and Diptera. Group VI was constituted by genes from beetles (Figure 4c).

The GH1 genes of different insect groups were distributed differently in GH1 groups, and the gene tree showed discordance with the species tree. All the genes from cockroaches and termites were clustered into a subgroup in Group I. The genes from hemipteroid groups were distributed into multiple subgroups in Group I and V. Most of the GH1 genes of Hymenoptera were clustered into one subgroup in Group V except that a few genes from *Harpegnathos*, *Polisters*, *Copidosoma,* and *Nasonia* formed another subgroup in the same group. The GH1 genes of beetles were distributed in Groups I, V, and VI, and the majority of identified beetle GHs were found in the latter two groups. GH1 genes in butterflies and moths were distributed into four clusters, with each cluster in Group I, II, III + IV, and V. The GH1 genes in flies and mosquitoes comprised two clusters, one in Group I and another in Group II.

### 3.4. Reconciliation between the Gene Tree and Species Tree

According to the inferred duplication history, we found massive duplications and losses in the multi-copy gene family in insects. The ancestors of insects and crustaceans had only one copy of GH1 (Figure 5a), while the ancestors of insects had duplicated it into four copies. A number of duplications were observed in the common ancestors of cockroaches and termites (6), thrips (11), and the ancestors of aphids (6). Large-scale duplications were found close to some tips of the beetle’s phylogeny (up to 40), as well as in the early branches of butterflies (15) and moths (20). Interestingly, we inferred a high number of duplications (up to 18) and losses (up to 24) during the evolution of Lepidoptera. Other orders also recorded a few losses and duplications, such as five gene losses in the common ancestors of Hymenoptera and four duplications in the common ancestors of parasitoid wasps and the ancestors of mosquitoes.

To determine if the duplications and losses were inflated by genome assemblies, we performed correlation analyses between both duplications and losses at the terminal branches and the BUSCO scores. As a result, we found low correlations between the losses and BUSCO missing scores (Figure 5b; R = 0.13, *p* = 0.41) and between the duplications and BUSCO duplicate scores (Figure 5c; R = 0.14, *p* = 0.36).

### 3.5. Collinearity and Duplication Modes

To understand how GH1 genes evolved in genomes, we analyzed the collinearity of the selected genomes in each order separately and inferred the duplication mode for all the genes in the selected species. We found that more than half of the identified GH1 genes in selected species (140 out of 239) were produced via tandem duplication (Figure 6).

The locations of the duplications differed in each insect group. In Hymenoptera, a few tandem duplications were recorded, where the duplication in *P. dominula* ess in a collinearity module with the gene in *B. terrestris* and *A. colombica* (Appendix A). In selected Coleoptera, 70 and 11 genes were tandem duplications and proximal duplications, respectively. A large group of duplications was found in a collinearity block between *T. castaneum* and *A. planipennis*, whereas the duplications of *A. glabripennis* and *O. taurus* were located outside of collinearity blocks (Appendix A). In selected Lepidoptera, 50 and 19 genes were tandem duplications and proximal duplications, respectively; the majority of tandem duplications were located on a few collinearity blocks, except a few duplications in *M. sexta* and *G. mellonella* (Figure 6). In addition, we found that five genes were segmental duplications in *S. frugiperda*. In Diptera, we observed tandem duplications (16 genes) present in all mosquitoes. The duplications in *C. quinquefasciatus* and *A. aegypti* were located in the collinearity blocks that contained the single GH1 gene of *D. melanogaster*; another duplication of *C. quinquefasciatus* was found in a collinearity block of *A. aegypti* and *C. quinquefasciatus* (Appendix A).

## 4. Discussion

β-glucosidase, mainly constituted by glycoside hydrolase family GH1, is a crucial enzyme in insect–plant interactions. After identifying GH1 genes from 54 selected arthropod species, we found that the GH1 gene is present in all insect species, whereas it appears to be lost in some crustaceans and all Chelicerata—another large group in Arthropoda. In insects, this gene family has an ancient origin and has undergone not only duplication and loss but also neofunctionalization.

GH1 is present in various kingdoms ranging from bacteria, and plants to animals [8,32]. The loss of GH1 in some arthropod groups might be due to the redundancy of GH1 in their adaptation to different environments, during which the function of GH1 in these species might be replaced by other GH families such as GH30 and GH9, the minority members of β-glucosidase [8,58]. Though the presence of bacterial GH1 genes in the mite genome suggests that their symbionts may also compensate for the role of GH1 in these groups, as the bacterial GH1 genes in moths are likely from *Enterobacter*, a group of gut bacteria [59,60] involved in the insecticide resistance of various moth species [60,61,62]. However, not all crustaceans lack GH1 genes; for instance, *Daphnia* has five copies of GH1 genes. Further investigation on how herbivores of some non-insect arthropods, such as *Tetranychus urticae*, utilize plant resources would help us understand their adaptation to plant feeding despite the loss of a key enzyme of ancient origin in their genomes [63,64].

In insects, the number of GH1 genes differs among orders. Such differences in GH1 gene numbers are related to their ecology (Figure 1), as the GH1 genes are involved in plant cell wall digestion and plant metabolite utilization [2,8,65]. Large gene numbers in Blattodea, Coleoptera, and Lepidoptera, mainly feeding on wood or leaves, indicates the need of GH1 for digestion or detoxification [2]; small gene numbers in Hymenoptera and Diptera, mainly feeding on limited plant cell wall components, such as nectar, pollen, or fruit, imply the redundancy of GH1 as in other non-herbivory insects. Furthermore, a convergent defense system between insects and plants could also contribute to the large number of GH1s because characterized glycoside-related defensive enzymes, including myrosinases and cyanogenic β-glucosidases, in both insects and plants, belong to GH1 [66]. These insects encounter a large number of metabolites while feeding and can sequester the defense molecules into the body for their defense instead of detoxifying them, which might also explain the large GH1 gene numbers in some species of Hemipteroid groups, namely aphids and thrips. These might also explain the relatively large GH1 gene number in daphnia and springtail; daphnia feeds on algae and cyanobacteria in the aquatic environment [67], and springtail feeds on plant organic materials or fungus in high organic soils or leaf litters [68].

As all insect GH1 genes originate from a common ancestor based on the gene phylogeny, duplications and losses of the insect GH1 gene family are likely associated with insect adaptations to the insect–plant interactions. The duplications of GH1 genes in the ancestors of all insects coincided with the ancient origin of insect–plant interactions [48,69]. Furthermore, a large part of duplications in beetles and the common ancestor of butterflies and moths appear to concur with the evolution of wood or plant feeding [52,53]. Furthermore, a continual loss of GH1 genes in hemipteroid except for duplications in aphids and thrips is consistent with the diversification of their feeding habitats [50] as that in Hymenoptera [49,70,71].

However, distinct patterns of duplication and loss may be related to the different evolution of herbivory in different insect orders. Massive duplications are present near the phylogeny tips of beetles [72], along with a few of them located in collinear blocks, suggesting that the evolution of GH1 genes in beetles is associated with the evolution of beetle–plant interactions, particularly the independent evolution of phytophagy [53], while in butterflies and moths, substantial duplications in their common ancestor followed by many duplications/losses during their evolution indicate a high turnover of GH1s and the dynamic interactions between Lepidoptera and plants required [52], corroborated by the finding that most GH1 are located in collinear blocks. Furthermore, a continual loss of GH1 genes in hemipteroid groups and Hymenoptera might also be related to their ecology, as the ancestors of both groups had close interactions with plants [49,50]. In bugs, the loss of GH1 could be the result of shifts in their ecology from herbivory to predation in their ancestors and from predation back to herbivory in subsequent lineages [50]. Similarly, in bees and ants, the loss of GH1 genes may also be associated with the evolution of their ecology shifting from herbivory to predation and/or from predation to herbivory [49,70,71]. After regaining herbivory, bees and ants adapted to different feeding materials, such as pollen and nectar, in which other carbohydrate digestion enzymes play the main roles [73]. However, duplications of GH1s in parasitoid wasps are possibly related to the adaptions to their hosts that are rich in plant secondary metabolites, but further evidence is needed [74]. In Diptera, the different patterns observed in flies and mosquitoes might be related to the different lifestyles of their larvae, the former feeding on various food resources and the latter feeding on aquatic algae or organic matter [75].

One fate of gene duplications is neofunctionalization. The GH1 gene family has various functions in insects [1,2,8]. The one copy of the GH1 gene inferred in the ancestor of insects indicates the gain of functions in GH1 genes during insect evolution. An important function is cellobiose digestion, which has been mainly studied in termites because of their high cellulose digestion ability [11,76,77,78,79]. The characterized digestive GH1s in different insects (Figure 3) were all clustered in groups I and V, which suggest the ancient origin of digestive function in insect GH1. The sex-specific GH1 gene characterized in termites was also found in Group I, indicating a gain of a new function of GH1 genes in communication at least in cockroaches and termites [22,24]. Another important function of GH1 in insects is detoxification, which helps insects to utilize the metabolites of plants and/or avoid the dual-component plant defense system [1,4,6,7,19,66]. The myrosinase of aphids was classified in Group I, indicating another gained new function of insect GH1 [14,27]. Moreover, the most similar sequence of identified myrosinase in beetles was found in Group VI, which corroborates that the myrosinases of beetles had evolved independently [18]. The last functionally characterized GH1, cyanogenic β-glucosidase, was found in the long branch of Group IV containing GH1 genes from butterflies and moths, suggesting a new function of GH1 was probably acquired in their ancestors. Although neo-functional enzyme activities were characterized in insect GH1 genes, including sex-specific glucosidase, myrosinase, and cyano β-glucosidase, the functions of GH1 in a large number of groups remain to be further studied, especially those groups with large duplications.

Another fate of duplication genes is to become pseudogenes and finally be lost during evolution. In our analysis, we noticed some GH1 genes only encode proteins with a partial domain of beta-glucosidase, especially in beetles and Lepidoptera. The partial domains are likely related to the loss of conserved functionally important sites, such as in some GH1s in Lepidoptera (Appendix A), which would incur non-functional proteins and indicate the pseudogenization of the duplicated GH1 genes. Further efforts are needed to explore these GH1 pseudogenes in insects, which might be related to the loss of GH1s in some arthropods.

Apart from the gene numbers, the expression levels of the GH1 genes and their expression locations in insects are also important. As tandem duplicates showed overactivated expression [80], more than half of the tandem duplications identified in GH1 genes suggest that the expression of the GH1 tandem cluster could affect insect adaptation to plant utilization. In addition, the GH1 gene family not only digests plant cell walls but also activate metabolites to form toxic compounds; therefore, it is important for insects to regulate the expression of the specifical enzyme in suitable tissues. The digestive beta-glucosides mainly function in the guts, where the physical environment, such as low pH, suppresses the enzymatic activity of defensive beta-glucosidase [6]; insect defensive beta-glucosidases were expressed in specific locations, for example, the sarcoplasm of non-flight muscle of aphids [14], defensive glands of leaf beetles [30], and hemocytes of *Zygaena filipendulae* larvae [31]. Upon attack, the insect could release the defensive beta-glucosidases to contact and activate the metabolites stored in separate locations for producing toxic products. Further studies at the gene expression level could help to reveal the role of the GH1 gene family in insect adaptation.

Overall, we analyzed the evolution of the key enzyme GH1 in insects. Our results showed that the GH1 genes in insects had ancient origins and experienced not only duplication and loss but also neofunctionalization. These evolutionary histories enriched the reservoir of GH1s for the digestion of plant cell walls as well as the utilization of plant defense. Further evolutionary analyses of other GH enzyme groups would help understand the loss of the GH1 gene in other arthropod groups, and the functional analyses of insect GH1s would provide more insights into insect–plant interactions.

## Figures and Tables

**Figure 1 insects-13-00786-f001:**
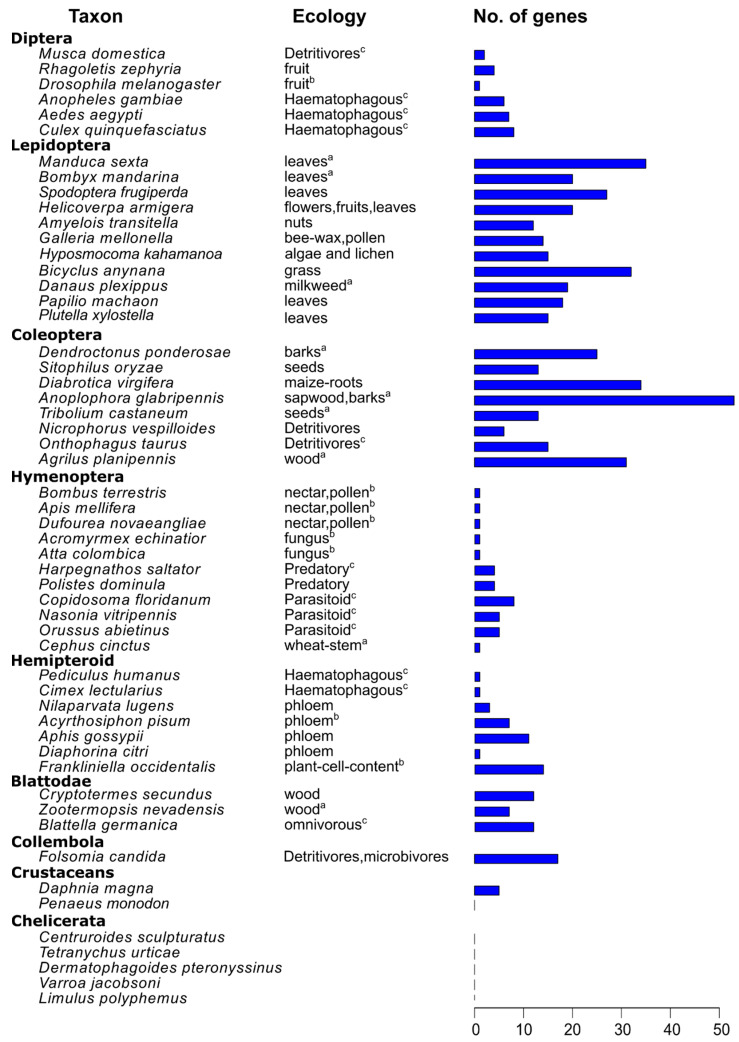
The feeding ecology and the number of identified GH1 genes in selected insects and other arthropods. Insect feeding ecology is indicated as their feeding types or feeding parts in plants. Blue bars represent the identified gene numbers. The species with labels were used for further phylANOVA analysis to compare the gene numbers between labeled groups, which was presented in Results; a: herbivores, direct feeding on plant cell walls; b: herbivores, non-direct feeding on plant cell walls; c: non-herbivores.

**Figure 2 insects-13-00786-f002:**
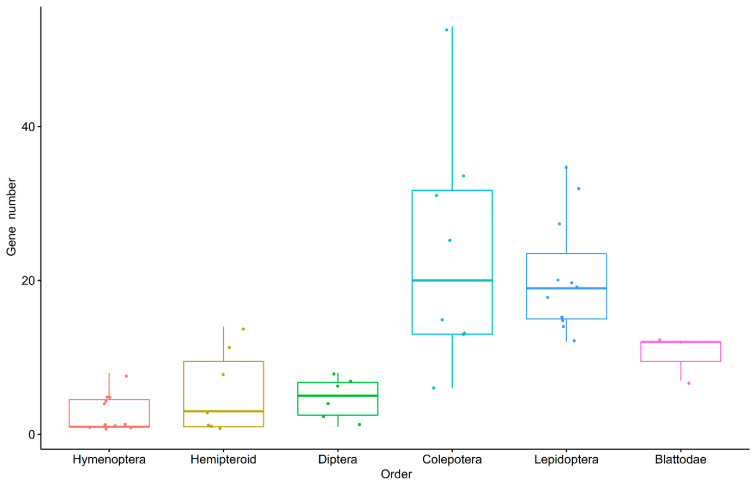
The GH1 gene numbers in each insect order.

**Figure 3 insects-13-00786-f003:**
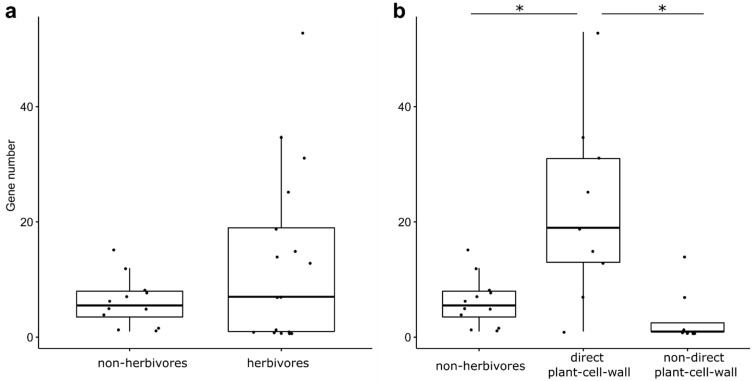
The GH1 gene numbers in different insect groups; * indicates significant differences between groups. (**a**) the comparison of GH1 gene number between non-herbivores and herbivores; (**b**) the comparison of GH1 gene numbers among non-herbivores and separated herbivory groups, including direct plant-cell-wall and non-direct plant-cell-wall.

**Figure 4 insects-13-00786-f004:**
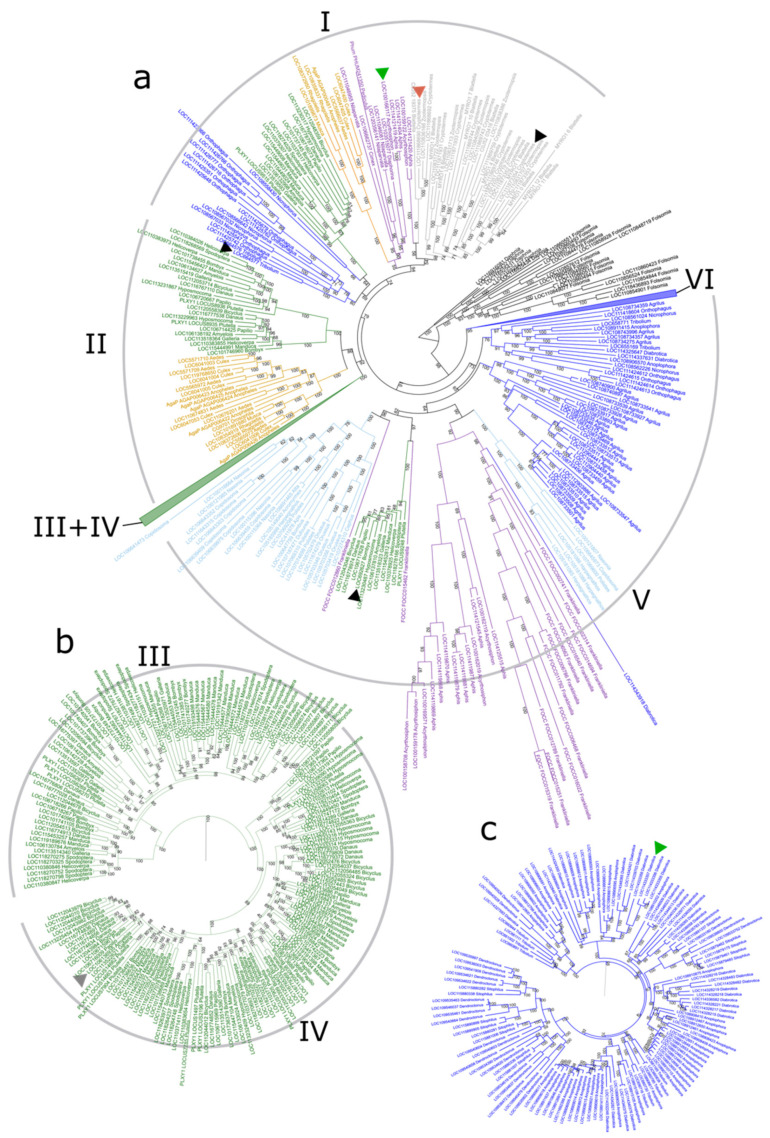
The phylogeny of identified GH1 genes built from proteins with IQTREE. The best blastp targets of characterized functional GH1s are marked in colored triangles. The GH1 gene trees were visualized using Interactive Tree Of Life (Letunic and Bork 2019). Digestive β-glucosidases (▲): LOC692627 *Bombyx* (AAP13852.1, blastp E-value: 0, (Byeon et al., 2005)); LOC111873352 *Cryptotermes* (AGS32242.1, blastp E-value: 0, (Franco Cairo et al., 2013)); LOC66457 *Tribolium* (AAG26008.1, blastp E-value: 0, (Ferreira et al., 2001)); sex-specific β-glucosidase (▲): LOC111868690 *Cryptotermes* (ABN05620.1, (Weil et al., 2007)); myrosinase (▲): LOC100166117 *Acyrthosiphon* (AAL25999.1, blastp E-value: 0, (Jones et al., 2002)); LOC114333357 *Diabrotica* (AHZ59651, blastp E-value:0, (Beran et al., 2014)); cyanogenic β-glucosidase (▲): LOC106136935 *Amyelois* (SGZ49382, blastp E-value:0, (Pentzold et al., 2017)). Bootstrap values were labeled at each node. The tip names contain gene id and genus name: (**a**) the whole phylogeny of all GH1 genes identified in this study with a collapsed clade of Group III + IV and a collapsed clade of Group VI. The colored branches represent genes from different groups: green: Lepidoptera, orange: Diptera, gray: Blattodea, blue: Coleoptera, purple: Hemipteroid, and turquoise: Hymenoptera; (**b**) the expanded clade of Group III and Group IV comprising GH1 genes from butterflies and moths; (**c**) the expanded clade of Group VI comprising GH1 genes from beetles.

**Figure 5 insects-13-00786-f005:**
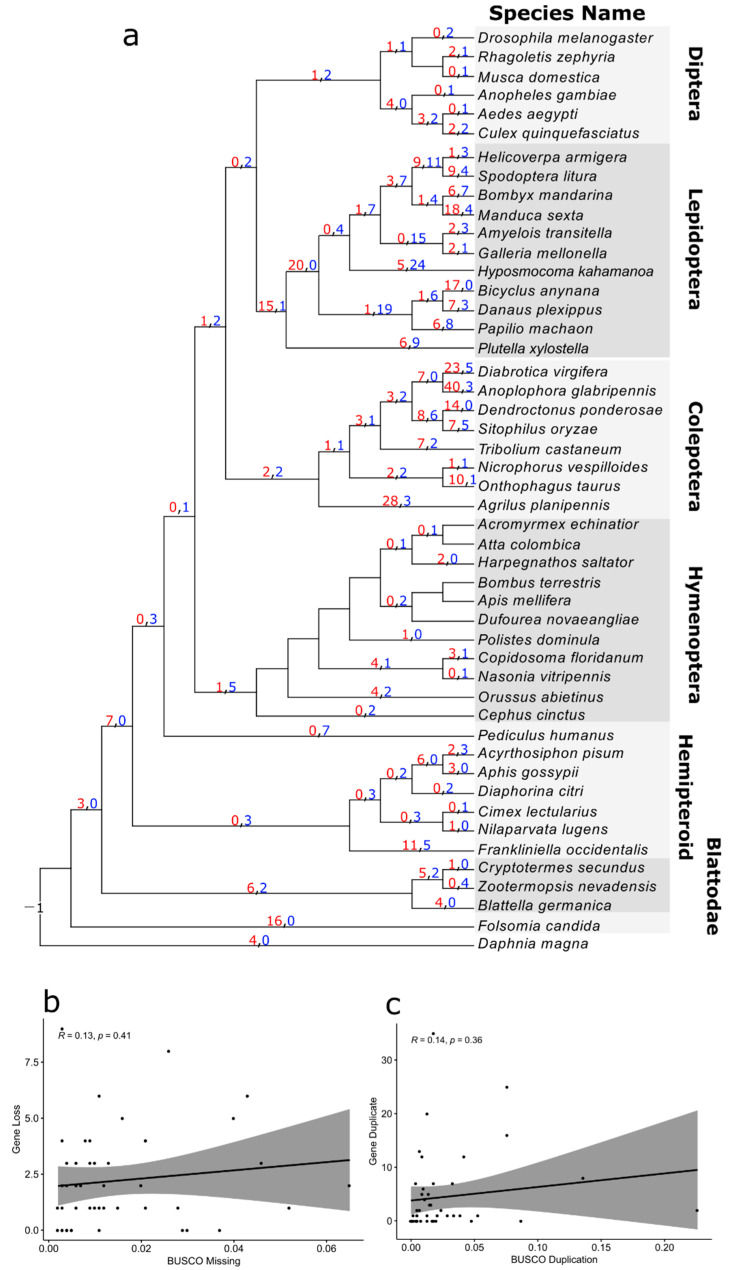
Duplication and loss inference via the reconciliation of the GH1 gene tree with a species tree: (**a**) the species tree was inferred from a few previous studies as described in the Methods section. Numbers in red indicate duplications, and numbers in blue indicate losses; (**b**) the correlation between inferred gene losses of terminal branches and BUSCO missing scores of the species at the corresponding tips; (**c**) the correlation between inferred gene duplications of terminal branches and BUSCO duplication scores of the species at the corresponding tips.

**Figure 6 insects-13-00786-f006:**
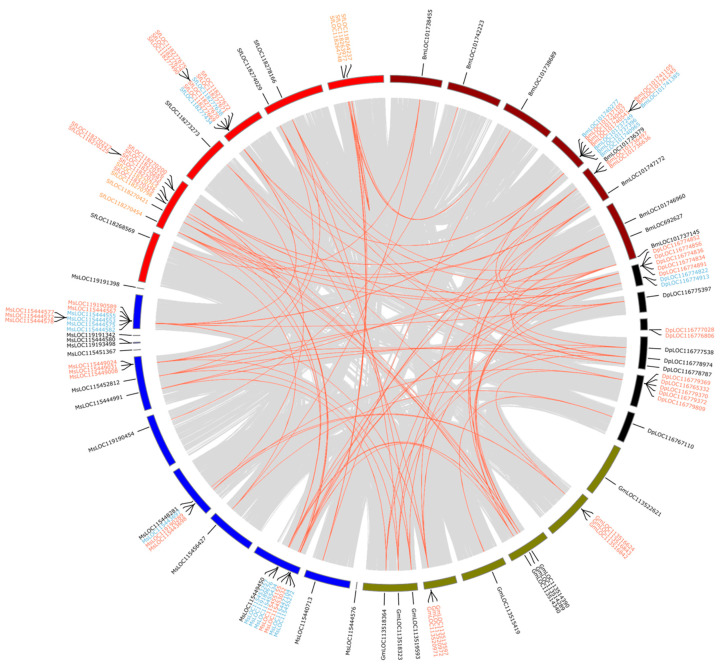
Chromosomal locations and duplications modes of identified GH1s in selected species of Lepidoptera. Gene names are presented with two-letter species abbreviations and gene ids; red, blue, black, and yellow gene names represent tandem duplications, proximal duplications, dispersed duplications, and segmental duplications, respectively. Different color of karyotype indicates different species and red lines link the GH1 genes from collinear blocks. Bm, *Bombyx mori*; Dp, *Danaus plexippus*; Gm, *Galleria mellonella*; Ms, *Manduca sexta*; Sf, *Spodoptera frugiperda*.

## Data Availability

All data associated with the study are publicly available.

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
