# Peer review of "The Evolution of Glycoside Hydrolase Family 1 in Insects Related to Their Adaptation to Plant Utilization"

_insects, 2022, doi:10.3390/insects13090786_

Round 1
Reviewer 1 Report
Dear Authors,
please, find detailed comments and suggestions added to the manuscript file.
In general, there are some parts that need to be shortened in the Results section. Moreover, several statements in the Discussion section are not supported by references. The entire manuscript contains some parts that are written two or three times just slightly different leaving the impression of lots of repetition. Please, put some effort in streamlining the manuscript to avoid these repetitive elements.
I would be happy to see a revised version of your manuscript.
Kind regards,

Author Response
Response to Reviewer 1 Comments
Point 1: In general, there are some parts that need to be shortened in the Results section. Moreover, several statements in the Discussion section are not supported by references. The entire manuscript contains some parts that are written two or three times just slightly different leaving the impression of lots of repetition. Please, put some effort in streamlining the manuscript to avoid these repetitive elements.
Response 1: We are grateful to the reviewer for reviewing our manuscript thoroughly and providing numerous constructive comments. We have tried our best to remove repetitive elements and reorganized several parts of the manuscript as well as revised the mentioned statements in the Discussion. With these revisions, we believe this manuscript has been significantly improved after incorporating these valuable suggestions.
Point 2: Line 29: Mannanases cannot hydrolyze cellulose.
Response 2: Mannanases indeed cannot hydrolyze cellulose but can hydrolyze hemicellulose. As the plant cell walls contain not only cellulose but also hemicellulose, pectin, and lignin, we changed “cellulose digestion enzymes” to “digestive enzymes”.
Point 3: Line 38-54: This whole paragraph has a big overlap with the paragraph before. I suggest to merge both paragraphs to avoid repetition of glucosidase functionalities.
Response 3: We thank the reviewer for the suggestion. After describing the two important adaptions of insect to plant utilization in the first paragraph, we further summarized the specific functions of β-glucosidases in the second paragraph. The second paragraph has a big overlap in the terms of the functions as glucosidases play an important role in insects’ utilization of plants. However, we believe it is necessary to introduce the functions of glucosidases since this work mainly focused on the evolution of glucoside hydrolase family 1, a large group of β-glucosidases. Therefore, we tried our best to minimize the overlap in the second paragraph.
Point 4: Line 64: This hypothesis implies that plants are a derived food source in insects and probably animals in general. I totally agree with this statement but I think it should be made clear to the reader! I suggest to add some paper like: doi.org/10.1002/evl3.127
Response 4: We thank the reviewer for the comment. We have changed the sentence with indicated reference in the revised manuscript. “It has been suggested that the insect GH1s diverged from the same animal GH1 gene ancestor and further evolved with functions in plant nutrient utilization (Ketudat Cairns and Esen 2010), which may be related to the evolution of plants as a derived food source of in insects(Román‐Palacios, Scholl et al. 2019).”
Point 5: Line 69-71: This sentence should be merged with line 53-54. The introduction needs some re-organization in general to make it less repetitive and easier to follow. It is clear to me what the authors want to highlight. However, it is not written in a direct way.
Response 5: As the reviewer suggested removing repetitions in the introduction and making it easy to follow, we re-organized this part and deleted several repetitive contents. In these revisions, the sentence in line 53-54 was deleted and this sentence was kept as our main point.
Point 6: Line 77: The fact that GH1s are encoded by a moderate-size gene family in insects needs to be mentioned earlier (in the general part of the introduction).
Response 6: It has been added in the revised introduction, as indicated in Line 58. “GH1s are encoded by a moderate-size gene family in insects.”
Point 7: Line 90: Which BUSCO gene set did you use? Please indicate.
Response 7: It has been added in the revised manuscript. “The completeness of the genome assemblies was determined by using BUSCO with insecta_odb10 dataset.”
Point 8: Line 96-97: I miss some of these classifications for certain taxa. Please, provide these details for each taxon if known. Additionally, I suggest to use color code instead of "a", "b" and "c" in the figure. Maybe this would work in combination with the bars indicating the number of genes i.e. color code these bars with e.g. green, red, blue instead of a, b and c.
Response 8: We apologize for the unclear statement. We had added the detailed feeding behavior for each species in Figure 1 as in the column of Ecology, which could indicate the feeding types we labeled. These classifications of “a”, “b”, and ”c”, were used for phyloANOVA analysis for comparing the gene numbers between these feeding types. The phyloANOVA analysis takes the phylogenetic background into account, and requires a dated species tree. As not all the species we analyzed were dated in previous studies, we only selected species that had dated in a previous study, as labeled, for the phyloANOVA analysis. Therefore, we also kept blue bars instead of colored bars. We also changed the figure caption to make it clearer. “The ecology of insects is indicated as their feeding types or feeding parts in plants. Blue bars represent the identified gene numbers. The species with labels were used for further phyloANOVA analysis to compare the gene numbers between labeled groups, which was presented in Figure 3; a: herbivores, direct feeding on plant-cell-walls; b: herbivores, non-direct feeding on plant-cell-walls; c: non- herbivores.”
Point 9: Line 158: As GH9 are widespread in insects like other metazoans, it would be interesting to compare/correlate their gene number as well as absence/presence with GH1s. You just need to have a look in your dbcan output files. Similarily, a correlation with cellulases absence/presence like GH45 would be of interest.
Response 9: We appreciate this constructive comment from the reviewer. It is fascinating, as the reviewer mentioned, to investigate the possible correlation of gene numbers or presence between these gene families in insect orders. As the interesting expansions and contractions in Lepidoptera and Coleoptera and the likely correlations to their dietary behaviors, we are currently analyzing all the reported carbohydrate-active enzyme gene families in both insect orders, including the correlation between these gene families, including GH45, GH9, and GH1, and their evolutions in much details. Our first preliminary result showed no correlation between the gene numbers of GH1 and GH9 in beetles.
Point 10: Line 190: This comparison may not be conclusive (or in other words just partially explain correlations seen in Fig.3) given the fact that GH1 proteins fulfill so many different functions as you pointed out in the introduction. I.e. a chewing herbivore may be exposed to many more secondary compounds released from its plant diet compared to a sucking herbivore.
Response 10: We partly agree with the reviewer on this comment. The GH1 proteins show many different functions, including digestion and detoxification. A chewing herbivore would be exposed to more secondary compounds compared with a sucking herbivore because of the release of metabolites during the destruction of plant cell walls. Therefore, we subsequently separated the plant feeding insects into two groups, direct plant-cell-wall feeding group, and non-direct plant-cell-wall feeding group, and compared the GH1 gene numbers between both groups. The result indeed shows a significant difference between both feeding groups as shown in Figure 3b.
Point 11: Line 197: There is only a single species representing this type of food source and it possesses 14 GH1 genes.
Response 11: We apologize for the unclear presentation of the comparison results. Indeed, we analyzed a single plant cell content feeding species, Frankliniella occidentalis in our study. Here we compared the gene numbers between species directly feeding on plant cell walls with species not directly feeding on plant cell walls, instead of using only one species for the comparison. We have changed the description to “Herbivores feeding directly on plant-cell-wall, including wood- and leaf- feeding, have significantly larger GH1 gene numbers than herbivores non-directly feeding on plant cell walls(t=5.15, p=0.006), such as phloem-sap or cell-content, and non- herbivores(t=4.82, p=0.006).”
Point 12: Line 205: I recommend to make use of the amino acid alignment beside phylogenetic tree reconstruction. Amino acid substitutions in functionally important sites may help to predict functionality of proteins.
Response 12: We appreciate the reviewer for the comment. As mentioned before, the moderate size of GH1 gene family contains around 570 genes, which makes it difficult to compare the functional sites in the alignment of all the genes. Therefore, we investigated the substitutions of functionally important sites of GH1 proteins in the lepidoptera, which contains a large number of genes and duplications and losses. The functionally important sites, including binding sites, catalytic sites, and other residues in the active center, were determined in previous studies (Ferreira, Marana et al. 2001, Byeon, Lee et al. 2005, Husebye, Arzt et al. 2005). As a result, we found that the important sites were highly conserved with only a few substitutions, while the residues next to these sites appear to contain much more substitutions (Supplementary Figure S2). This suggests that the conformations of the active centers instead of the functional sites may be related to the diversified functions (Ketudat Cairns and Esen 2010). In addition, we observed that some genes lacked part of the conserved functional sites, which indicated the pseudogenization of these genes. We have added this information in the revised manuscript as the reviewer suggested, in Line 394-397 in the Discussion part. “The partial domains likely are related to the loss of conserved functionally important sites, like in some GH1s in Lepidoptera (Supplementary Figure 2), which would incur non-functional proteins and indicate the pseudogenization of the duplicated GH1 genes.”
Point 13: Line 216: Please, enlarge the triangles indicating functionally characterized proteins as they are hard to see.
Response 13: We thank the reviewer for the suggestion. The triangles have been enlarged in the updated Figure 4.
Point 14: Line 227: This should be also indicated in panel a) of the figure, like it is in case of VI.
Response 14: It has been added to panel a) of the figure.
Point 15: Line 232: indicating massive gain (duplication) and loss events in this multi-copy gene family.
Response 15: It has been added in Line 239-240. “According to the inferred duplication history, we found massive duplications and losses in the multi-copy gene family in insects.”
Point 16: Line 241: First observation: discordance between gene tree and species tree.
Response 16: This information has been added to Line 229. “The GH1 genes of different insect groups were distributed differently in GH1 groups and the gene tree showed a large discordance with the species tree.”
Point 17: Line 244-253: This part needs to be shortened as it contains nothing but numbers and taxa and those were not correlated with the evolution of GH1 functionalities.
Response 17: We thank the reviewer for this comment, but we are afraid that we do not completely agree with the reviewer on this point. This part indeed contains a lot of numbers and taxa. However, these duplications and losses are possibly involved in the evolution of herbivory in insects. The duplications of GH1 genes in beetles and butterflies and moths appear to coincide with the evolution of wood or plant feeding. Besides, a continual loss of GH1 genes in hemipteroid and Hymenoptera is likely related to the diversification of their feeding habitats. Furthermore, the varied numbers on the internal branches of insect orders indicate the different evolution scenarios of the GH1, which possibly were involved in the evolution of herbivory in different orders. Thus, we tried our best to shorten this paragraph but kept the interesting numbers in the revised text.
Point 18: Line 266: This part needs to be shortened as it contains nothing but numbers and taxa and those were not correlated with the evolution of GH1 functionalities.
Response 18: We thank the reviewer for this comment, but we do not totally agree on this point. The numbers and taxa in this part might be related to the evolution of GH1 functionalities. A large number of the GH1 duplications we found in this study were derived from single gene duplication except a few in S. frugiperda. Recurrent tandem gene duplication, as we found in GH1 gene family, could contribute to the divergent gene functions under positive selections or tissue specific expression in insects (Fan, Chen et al. 2008, Dai, Kiuchi et al. 2021). At the same time, the different modes of duplications also have been related to divergent evolutionary patterns and contribute differently to the expansion of gene families (Qiao, Yin et al. 2018). Although we didn’t relate the duplication modes to the functions of GH1s, we hope it could be useful for further studies on the evolution of GH1 gene families, such as the evolution of tissue specific expression of functional GH1 genes. Therefore, we shortened this part and kept the interesting description in the revised text.
Point 19: Line 277: This figure is far too small to see any detail. I am aware of the problem with visualizing big data sets. However, you need to find a way that one can at least read the gene names. I suggest to show just a single panel or two as representatives instead of all the four panels.
Response 19: We appreciated the reviewer for the suggestion. As suggested, we kept a single panel with the most interesting order-Lepidoptera-in the figure and put the other three in Supplementary Figure S1.
Point 20: Line 305: What about Myriapoda, another arthropod taxon?
Response 20: We have checked the presence of GH1 gene family in the 9 available Myriapoda genomes in the NCBI Genome database by tblastn with the identified GH1s in Daphnia Magna as query input. As a result, 5 out of 9 genomes have no significant queried target, and the targets in the other 4 genomes, including Glomeris maerens, Platydesmidae sp., Julidae sp., Strigamia maritima, showed low similarity. After manually checking the 3 most significant targets on dbCAN2 online server, we were not able to identify them as carbohydrate active enzyme. Based on these results, we believe that no GH1 gene is present in Myriapoda as well.
Point 21: Line 309: Repetition of line 305.
Response 21: The sentence has been merged with the next sentence in the paragraph. “The loss of GH1 in some arthropod groups might be due to the redundancy of GH1 in their adaptation to different environments, during which the function of GH1 in these species might be replaced by other GH families such as GH30 and GH9, the minority members of β-glucosidase (Ketudat Cairns and Esen 2010; Kao, et al. 2016).”
Point 22: Line 312-313: Are there GH9 or GH30 in Chelicerata?
Response 22: Yes, we found GH9 and GH30 in all the studied Chelicerata species.
Point 23: Line 314-315: Are there any bacterial symbionts of mites known?
Response 23: According to a review summarizing the bacterial flora in mites, Cardinium, Orientia, Wolbachia were the dominant bacteria studied in mites, while Rickettsia, Anaplasma, Bartonella, Francisella, Coxiella, Borrelia, Salmonella, Erysipelothrix and Serratia were also reported (Qiao, Yin et al. 2018). A study focused on the bacterial communities of stored product mites has shown a similar composition(Hubert, Nesvorna et al. 2021). In addition, a study on Varroa destructor in honey hives showed the presence of mite-specific Diplorickettsia as well as other shared bacteria, including Arsenophonus, Morganella, Spiroplasma, Enterococcus, and Pseudomonas with their host honey bees(Hubert, Kamler et al. 2016).
Point 24: Line 318-Line 319: Again, this is a sucking species that may not need to a) digest much of cellulose, if GH1 is involved in cellulose digestion and b) deal with many plant secondary compounds.
Response 24: We totally agree with the reviewer on the point. The sucking spider mites may not need various functional GH1 as other sucking insects, but enzymes are needed to deal with even a small number of secondary metabolites. It is, therefore, still interesting to know why this ancient gene family has been totally lost in this species and probably in other plant feeding arthropods.
Point 25: Line 324-329: This argument is not justified. I do not see a reason for massive gene duplication in GH1 to digest a simple substrate like cellobiose or triose. One or two duplicates may persist to increase gene dosage but apart from that it is more likely that duplicated genes persisted due to sub- or neo-functionalization.
Response 25: We thank the reviewer for pointing this out. Indeed, the mass duplications of GH1 genes likely are related to the acquired functions instead of only digestion. According to the gene phylogeny, a large number of duplicated genes were clustered with characterized GH1 genes related to defense in Coleoptera and Lepidoptera. Therefore, we revised the sentence to “Large gene numbers in Blattodea, Coleoptera, and Lepidoptera, mainly feeding on wood or leaves, indicates the need of GH1 for digestion or detoxification (Tokuda 2019); small gene numbers in Hymenoptera and Diptera, mainly feeding on limited plant cell wall components, such as nectar, pollen, or fruit, imply the redundancy of GH1 as in other non-herbivory insects.”
Point 26: Line 335-339: I would not restrict myself to myrosinases, as they make just one or two copies. It is plausible to hypothesize that aglucone diversity drives GH1 diversification, or?
Response 26: We thank the reviewer for the comment. We have changed the myrisonases to glycoside-related defensive enzymes. “Furthermore, a convergent defence system between insects and plants could also contribute to the large number of GH1s because characterized glycoside-related defensive enzymes, including myrisonases and cyanogenic β-glucosidases, in both insects and plants belong to GH1 (Bhat and Vyas 2019; Pentzold, et al. 2014).” We also agree with the hypothesis that the reviewer proposed, but we think it is not well supported without further experimental data on GH1 enzymatic activities with related secondary metabolites.
Point 27: Line 350-354: I do not get this point.
Response 27: We apologize for the unclear description. We have re-written this sentence. “Furthermore, a large part of duplications in beetles and in the common ancestor of butterflies and moths appear to concur with the evolution of wood or plant feeding (McKenna, et al. 2019; Kawahara, et al. 2019).”
Point 28: Line 362: “phytophagous” to “phytophagy”
Response 28: We thank the reviewer for spotting this error. It has been corrected.
Point 29: Line 411-416: Here you may add the information from the GH1 multiple amino acid alignment. Are there any GH1 proteins accumulating amino acid substitutions in functionally important sites?
Response 29: We thank the reviewer for the comment. We have addressed this comment as the comment of Line 205.
Reference
Byeon, G. M., K. S. Lee, Z. Z. Gui, I. Kim, P. D. Kang, S. M. Lee, H. D. Sohn and B. R. Jin (2005). "A digestive β-glucosidase from the silkworm, Bombyx mori: cDNA cloning, expression and enzymatic characterization." COMP BIOCHEM PHYS B 141(4): 418-427.
Dai, X., T. Kiuchi, Y. Zhou, S. Jia, Y. Xu, S. Katsuma, T. Shimada and H. Wang (2021). "Horizontal gene transfer and gene duplication of β-fructofuranosidase confer lepidopteran insects metabolic benefits." Molecular Biology and Evolution 38(7): 2897-2914.
Fan, C., Y. Chen and M. Long (2008). "Recurrent tandem gene duplication gave rise to functionally divergent genes in Drosophila." Molecular biology and evolution 25(7): 1451-1458.
Ferreira, A. H. P., S. R. Marana, W. R. Terra and C. Ferreira (2001). "Purification, molecular cloning, and properties of a β-glycosidase isolated from midgut lumen of Tenebrio molitor (Coleoptera) larvae." INSECT BIOCHEM MOLEC 31(11): 1065-1076.
Hubert, J., M. Kamler, M. Nesvorna, O. Ledvinka, J. Kopecky and T. Erban (2016). "Comparison of Varroa destructor and worker honeybee microbiota within hives indicates shared bacteria." Microbial ecology 72(2): 448-459.
Hubert, J., M. Nesvorna, S. J. Green and P. B. Klimov (2021). "Microbial communities of stored product mites: variation by species and population." Microbial ecology 81(2): 506-522.
Husebye, H., S. Arzt, W. P. Burmeister, F. V. Härtel, A. Brandt, J. T. Rossiter and A. M. Bones (2005). "Crystal structure at 1.1Å resolution of an insect myrosinase from Brevicoryne brassicae shows its close relationship to β-glucosidases." INSECT BIOCHEM MOLEC 35(12): 1311-1320.
Ketudat Cairns, J. R. and A. Esen (2010). "β-Glucosidases." CELL MOL LIFE SCI 67(20): 3389-3405.
Qiao, X., H. Yin, L. Li, R. Wang, J. Wu, J. Wu and S. Zhang (2018). "Different modes of gene duplication show divergent evolutionary patterns and contribute differently to the expansion of gene families involved in important fruit traits in pear (Pyrus bretschneideri)." Frontiers in Plant Science 9: 161.
Reviewer 2 Report
line 26 : "speciose" meant diversified ? "adapted" and not adapt
line 28 : "Most insects are herbivore and utilize plants …" Not "They" because not all insects are phytophagous. line 31 : (Rane, et al. 2019) after metabolites line 68 : indicated line 79 : revealed Descriptors should be added in Figure 1. Line 159_161 : to be deleted, this is material and method description. Latin names for gene should be abbreviated Results : F and t values with 2 decimals are enough lines 291and 295 : were and not are line 298: contained and not containAuthor Response
Response to Reviewer 2 Comments
We thank the reviewer for reviewing our manuscript and providing useful comments.
Point 1: line 26 : "speciose" meant diversified ? "adapted" and not adapt
Response 1: Speciose means species rich, comprising many species. The “adapt” has been corrected as “adapted” in manuscript.
Point 2: line 28 : "Most insects are herbivore and utilize plants …" Not "They" because not all insects are phytophagous.
Response 2: We thank the review for the correction, which has been implemented in the revised text.
Point 3: line 31 : (Rane, et al. 2019) after metabolites
Response 3: It has been corrected.
Point 4: line 68 : indicated
Response 4: It has been corrected.
Point 5: line 79 : revealed
Response 5: It has been corrected.
Point 6: Descriptors should be added in Figure 1.
Response 6: It has been added as “The feeding ecology and the number of identified GH1 genes in selected insects and other arthropods.”
Point 7: Line 159_161 : to be deleted, this is material and method description.
Response 7: It has bene deleted.
Point 8: Latin names for gene should be abbreviated
Response 8: We have thoroughly checked the Latin names for genes in the wholel manuscript.
Point 9: Results : F and t values with 2 decimals are enough
Response 9: They have been corrected.
Point 10: lines 291and 295 : were and not are.
Response 10: It have been corrected.
Point 11: line 298: contained and not contain
Response 11: It has been corrected.
Round 2
Reviewer 1 Report
Dear Authors,
thank you for taking the review process serious!
You provided a conclusive and clear point-by-point response.
Kind regards